# The Novel Role of Phage Particles in Chronic Liver Diseases

**DOI:** 10.3390/microorganisms11051181

**Published:** 2023-04-30

**Authors:** Liuying Chen, Xiaohua Hou, Huikuan Chu

**Affiliations:** Division of Gastroenterology, Union Hospital, Tongji Medical College, Huazhong University of Science and Technology, 1277 Jiefang Avenue, Wuhan 430022, China

**Keywords:** virus, phage, chronic liver disease

## Abstract

The gut microbiome is made up of bacteria, fungi, viruses and archaea, all of which are closely related with human health. As the main component of enterovirus, the role of bacteriophages (phages) in chronic liver disease has been gradually recognized. Chronic liver diseases, including alcohol-related liver disease and nonalcoholic fatty liver disease, exhibit alterations of the enteric phages. Phages shape intestinal bacterial colonization and regulate bacterial metabolism. Phages adjoining to intestinal epithelial cells prevent bacteria from invading the intestinal barrier, and mediate intestinal inflammatory response. Phages are also observed increasing intestinal permeability and migrating to peripheral blood and organs, likely contributing to inflammatory injury in chronic liver diseases. By preying on harmful bacteria, phages can improve the gut microbiome of patients with chronic liver disease and thus act as an effective treatment method.

## 1. Introduction

The phage is small and has a tadpole shape, microsphere shape and fine rod shape. Phages are composed of genomic nucleic acids and capsid proteins, and nucleic acids can be linear double-stranded DNA, circular single-stranded DNA, or linear single and double-stranded RNA [1]. With the development of large-scale viral metagenomics, we have gained an insight into the diversity of phages. Some phages without tails are being recognized; some phages present an outer lipid membrane in addition to their protein capsid. Most phages in human intestinal flora belong to the order *Caudovirales*, with double-stranded DNA smaller than 200 KB in size. Megaphage with a very large genome (540 KB in length) has been found in animal and human gut [2].

Phages can dissolve bacteria to complete their life cycle [3]. Gentle survival strategies are also used, some of which are even beneficial to bacteria. For example, temperate phages steadily integrate into bacterial genomes and confer antibiotic resistance to the host. Phages can be classified as either lytic or lysogenic depending on the types of interaction with their host [4] (Figure 1). In lytic growth, phages infect bacteria and synthesize their own components, which are assembled into new particles, thus releasing new virions by lysing host bacteria. During lysogenic growth, phages inject their genomes into the bacteria and integrate them into the bacterial chromosomes, replicating through host cell division. When the phages enter the bacteria body but remain in a static state, it is called pseudolysogeny [5]. After receiving the induction signal, the phages can reproduce by lytic or lysogenic methods. Finally, some phages, such as filamentous phages, can infect bacteria and then bud to produce new phages, which can save the host bacteria from lysis and death [6].

Virions are unique and can be replicated to a certain extent. The phage populations of patients with fecal microbiota transplantation (FMT) showed highly-donor-similar characteristics, and the diversity and abundance of the virus populations were comparable to that of the donors [7]. No significant changes in bacterial alpha and beta diversities of healthy adults treated with bacteriophages for 28 days in contrast to those treated with a placebo were observed, but an increase in members of the butyrate-producing *Eubacterium* and a decrease in taxa of *Clostridium perfringens* were found in the bacteriophages treatment group [8].

In recent years, a growing number of works have revealed the effects of enteric colonizing viruses on human health and disease, including chronic liver disease [9]. The role of enteroviruses in the progression and prognosis of chronic liver disease has been further understood, and some phages have been used as treatment for chronic liver disease [4]. By preying on bacteria, phages participate in the microecological balance of the gut and affect human health. Phages have been reported regulating the immune microenvironment of the intestine, and affecting blood glucose homeostasis [10,11]. The ecological balance between phages and bacteria is key to maintaining health. Our understanding of how phages directly or indirectly affect human health and chronic liver diseases is relatively limited.

## 2. Alterations in the Enteric Phages of Chronic Liver Diseases

### 2.1. Alcoholic Liver Diseases

Patients with alcohol-use disorder were reported with decreased abundance of *Propionibacterium*, *Lactobacillus* and *Leuconostoc* phages, and increased abundance of *Streptococcus* and *Lactococcus* phages compared with controls, whereas after 2 weeks of alcohol abstinence, all phages were increased [12]. Alcohol-use disorder patients with progressive liver disease were more abundant in *Enterobacteria* and *Lactococcus* phages than nonprogressive individuals. Patients with alcoholic hepatitis had the highest viral diversity and richness, compared with controls and patients with alcohol dependence [13].

Alcoholic hepatitis patients were dominated by *Lactobacillus*, *Streptococcus* and *Escherichia* phages, and alcohol-use disorders were mainly characterized by *Lactococcus* phages. When compared with controls, alcoholic hepatitis patients had decreased abundances of *Lactococcus* and *Parabacteroides* phages, and increased abundance of *Lactobacillus*, *Escherichia*, *Enterobacteria* and *Enterococcus* phages. The *herpesviridae* was exclusively detected in stool samples from patients with alcoholic hepatitis, and not found in patients with alcohol-use disorders and controls [13,14].

Alcoholic hepatitis with high MELD-score (>26.3) had the most abundant T4 virus, *Staphylococcus* phages, *Phietavirus* and *Citrobacter* phages, compared with the medium (between 22.1 and 26.3) and lower MELD-score patients (<22.1). Cox regression analysis showed Yersinia phages and Streptococcus phage Abc2 had 1.52 and 1.27 hazard ratios, respectively, for 90-day survival of alcoholic hepatitis patients. On the other hand, phages also have a therapeutic effect. Phages targeting *Enterococcus faecalis* provide a new therapeutic approach for alcoholic hepatitis [15].

These results suggest that phage changes are closely related to the development and prognosis of alcohol-related liver disease, and some phages may be used as prognostic markers and therapeutic targets.

### 2.2. Nonalcoholic Fatty Liver Disease

Experimental nonalcoholic liver disease animals were found with changes of intestinal virus. The short-term high-fat diet (16 weeks) increased the alpha diversity of viromes in mice, but the long-term high-fat diet (28 weeks) restored the alpha diversity of viromes identically to the control group [16]. What is more, Eukaryotic viruses *Phycodnaviridae* and *Mimivirdae* gradually replaced *Siphoviridae* in the dominant position of mice intestines during feeding with a high-fat diet. Viral communities of mice with a high-milk-fat diet were sensitive to dietary disturbances, in contrast to mice with baseline low-fat diet [17]. In mice fed a low-fat diet, the components of the mucosal viromes were significantly different from those of the luminal, but this difference disappeared in high-fat and high-sucrose-fed mice [18]. Furthermore, the abundance of *Caudovirales* phages obviously increased in the intestinal mucosa and lumen of high-fat and high-sucrose-fed mice compared with the control mice.

Anyway, gut virus is also related to the severity of NAFLD. The viral diversity was very different among patients with nonalcoholic fatty liver disease (NAFLD) and different disease stages, and controls. Patients with NAFLD activity score (NAS) 5–8 or cirrhosis had a significantly lower viral diversity and decreased phages proportions compared with NAS 0–4 patients and controls [19]. *Lactococcus* and *Streptococcus* phages were more abundant in NAS 5–8 or cirrhosis patients than NAS 0–4 patients and controls. Meanwhile, *Lactococcus* and *Leuconostoc* phages were significantly decreased, and *Lactobacillus* phage phiAT3 was significantly increased in NAFLD patients with higher degrees of liver fibrosis (F2–F4), compared with those with no or minimal liver fibrosis (F0–F1). The authors also built models that computed with age and AST, or computed with age, AST and platelet counts, to noninvasively detect the presence of NAS 5–8 or cirrhosis and F2–F4 liver fibrosis, respectively, while adding the viral diversity but not the bacterial diversity to the models improved the diagnostic accuracy.

### 2.3. Liver Cirrhosis

Relationship of disease progression of liver cirrhosis and gut virus was sparse. Jasmohan S Bajaj et al. reported compensated liver cirrhosis had similar alpha diversity of phage genera with controls [20]. Beta diversity analysis at the levels of phage families and phage genera showed liver cirrhosis patients were clustered. Furthermore, lactulose and rifaximin treatment of cirrhosis with hepatic encephalopathy had no effects on the intestinal phage genera. Correlation networks between phages and bacterial species were significantly reduced in compensated cirrhosis compared with controls. *Faecalibacterium*, *Streptococcus*, *Lactobacillus*, *Microbacterium* and *Lactococcus* phages were obviously increased in patients with cirrhosis, and *Kagunavirus* phage was decreased, compared with healthy controls. *Bullavirinae*, *Felixounavirus*, *Streptococcus*, *Escherichia* and *Pseudomomas* phages were positively related while *Faecalibacterium* phages were negatively related with the model for end-stage liver disease (MELD) scores.

## 3. Underlying Mechanisms of Phages in Chronic Liver Diseases

At present, the mechanism of phage in liver injury has not been studied deeply. The complex relationship between phages and bacteria makes the work tricky. Phages can play a role in chronic liver disease through known mechanisms, such as the increase in harmful bacteria and toxic substances, by altering the composition and metabolic capacity of gut bacteria. Phages also break through the intestinal barrier, cause inflammatory responses in the liver. Phage-caused intestinal inflammation can further aggravate damage of the intestinal barrier and promote the migration of bacterial and pathogen-associated molecular patterns (PAMPs).

### 3.1. Phages Shape Intestinal Bacterial Colonization

Bacteriophages are viewed as “commensal” gut-resident viruses, regulate bacterial composition and help maintain a “healthy” microbiota status [21,22]. How do bacteria live with deadly phages? The mutual cycle of resistance evolution and infection resistance evolution provides the theoretical basis for the coexistence of bacteria and phages [23,24]. Recent studies have shown that the physical environment in which bacteria and phages reside plays an important role in promoting bacteria-phage coexistence [23]. Researchers have shown that the coexistence of bacteria and phages does not depend on the development of anti-phage bacterium, nor on the phage’s ability to extend its host range. In contrast, the areas of mucous membranes that are inaccessible to phages are habitats for bacteria [25].

The influence of phages on bacterial evolution is universal; about 2 × 10^16^ phage-mediated gene transfer events occur every second [26]. Through transformation, generalized and specialized transduction and chromosomal rearrangements, phages transfer DNA into bacterial genome, and provide virulence genes to bacteria. The Vibrio cholerae toxin gene was derived from the filamentous phage CTXΦ(25) [27]. Phages encoding virulence factors are involved in the emergence of new epidemic strains of Salmonella [28]. By trapping iron, phages can affect the virulence of intestinal bacterial pathogens such as Vibrio vulnificus, Salmonella typhimurium, and Yersinia [29]. Phage-mediated exchange of resistance gene islands among enterococci provides a survival advantage for enterococci in the face of complex environmental factors [30]. Phage has been found hitchhiking on carrier bacteria to facilitate its infection of host bacteria, and promote carrier bacteria colonization [31]. In general, phages target the invading bacteria; sometimes phages cannot completely eradicate target bacteria [32]. Instead, they coexist with the target bacteria, affect non-susceptible species in the intestinal bacterial community by cascading effects and ultimately regulate the gut metabolome.

Phages from different taxa with different lifestyles can infect the same host, which may lead to within-host phage–phage interactions [33]. To compete for host cell resources, many phages have evolved superinfection exclusion mechanisms that enable the host’s resistance to repeated infection by other phage. Pf phages in *P. aeruginosa* can exclude infection of other phages by interfering with the function of type IV pili (T4P), which may reduce the efficacy of phages in treating pseudomonas-associated infections [34]. *Escherichia coli* abolishes further infection by bacteriophage T5 via the jamming of TonB-dependent transporters by small phage lipoproteins [35]. Host ecology and the interaction between the surrounding viruses are the two main factors that determine the pattern of viral co-infection [36].

The interactions of phage and bacteria are very complicated. Under great pressure from phage invasion, bacteria have evolved various immune mechanisms to resist the invasion. Meanwhile, phages also mount numbers of counter-defense strategies in this race for survival [37]. Bacteria develop resistance mechanisms to block phage infection in each step of attachment, genome injection, replication and cleavage [38]. For example, bacteria use elements related to phages such as R-type pyocins and type VI secretion systems (T6SSs) as part of their defense system; both are similar to the structure of phage tail [39]. In the presence of competing strains, fecal *Escherichia coli* V583 produced the phage ϕV1/7 to establish and maintain a dominance of its intestinal niche [40]. Bacteriophage infects the *Enterococcus faecalis* dependent on, the enterococcal polysaccharide antigen (Epa). Exposure to phages promotes mutations in the *Enterococcus faecalis* Epa gene, resulting in loss of resistance to cell-wall-targeting antibiotics and inadequate intestinal colonization [41,42]. In response to the SOS stress, *Lactobacillus reuteri* produces a significant number of phages, which provides it with a competitive advantage in the gut [43]. A commercial bacteriophage cocktail (against *Enterobacteriaceae*, *Staphylococcaceae*, and *Streptococcaceae* families) challenge alters intestinal bacterial communities, illustrated as decreased abundance of *Blautia*, *Catenibacterium*, *Lactobacillus*, and *Faecalibacterium* genus and increased abundance of *Butyrivibrio*, *Oscillospira* and *Ruminococcus* genus [44].

Most studies have shown changes in intestinal phage composition of patients with chronic liver disease, but the causes and consequences of these changes are rarely discussed. Depletion of certain probiotics such as *bifidobacterium* species and their phages in chronic liver disease might represent a common mechanism. The study of bacteria-phage interaction can better reveal the role of different phage abundance changes in chronic liver disease. *Bacteroides* phage BV01 was found inhibiting bile acid deconjugation, a mechanism by which the phage may influence hepatic disease, as microbially modified bile acids are important signals for hepatic fat metabolism [45].

### 3.2. Phages Regulate Bacterial Metabolism

With advances in genome sequencing, phages are found altering the metabolism of their hosts by carrying auxiliary metabolic genes (AMGs) [46]. Phages were suggested help digest plant-derived polymers by encoding the CAZyme gene which frees beta-1, 4-linked mannoses from galactomannan and glucomannan [47]. Phages prey on harmful bacteria and can eliminate their harmful metabolites [48]. Tryptophan is decarboxylated by gut bacteria to produce tryptamine, an important neurotransmitter. Phage (T4 and F1) intervention resulted in a significant decrease in the abundance of *C. sporogenes* in the intestinal tract of mice, along with a decrease in the content of tryptamine [32]. Germfree (GF) mice that were colonized by a ten-member bacterial community treated with phages T4 and F1, had significant changes in 17% of Fecal metabolites representing nearly all the KEGG pathways (e.g., amino acids, peptides, carbohydrates, lipids and so on) [32]. After further intervention with phage VD13 and B40-8, another 0.7% of metabolites were significantly affected. It is well established that phages can cause changes in intestinal metabolites, but the role of these changes is unclear. Changed bacterial metabolites caused by phages in liver diseases need further study.

### 3.3. Phages and Intestinal Barrier

Numerous studies have shown that phages can translocate directly through the mucosal barrier [49] (Figure 2). Phages administered orally, intranasally and intraperitoneally can appear rapidly in the blood and accumulate in the kidneys, spleen, liver and thymus [50]. Rats treated with a bacteriophage cocktail for 10 days showed increased intestinal permeability [51]. The mechanism of phage penetration of the mucosal barrier is still not completely clear. There are several hypotheses: phage uptake by epithelial cells through transcytosis; phages hidden in bacteria cross the epithelial barrier through the Trojan horse mechanism; phages enclosed in intestinal contents and ingested by enteric dendritic cells; or phages pass through damaged epithelial barriers [52]. High levels of phages were found in the mucous layer of the gut. Phages use their Ig-like domain proteins to bind to the variable glycan residues that coat the mucin glycoprotein, and colonize the intestinal mucous layer [53]. Phages in close proximity to intestinal epithelial cells may be a previously unrecognized antimicrobial defense mechanism that prevents bacteria from invading the intestinal barrier, and actively protects mucosal surfaces [54]. However, bacteriophages can also induce increased intestinal permeability [44,51]. Disruption of intestinal barrier integrity was found in rats ten days after administration of a bacteriophage cocktail active against *Staphylococcus*, *Streptococcus*, *Proteus*, *Pseudomonas*, *E. coli*, *K. pneumonia*, and *Salmonella*. Phages can also penetrate epithelial cell layers and migrate to peripheral blood and organs, likely contributing to inflammatory injury in chronic liver disease [55]. The relationship between phage and the intestinal barrier deserves further study.

### 3.4. Phages and Intestinal Inflammation

Bacterial components and phage DNA, released from the lysed bacteria, are the main ligands that trigger intestinal immunity through pattern recognition receptors (PRRs) in epithelial cells or immune cells (Figure 2). In addition, bacteriophages can directly motivate inflammatory responses in epithelial cells and immune cells in the intestine [56]. Phages have immunomodulatory activity, affecting the function of major populations of immune cells involved in innate and adaptive immune responses including phagocytosis and respiratory burst of phagocytes, cytokine production and production of antibodies against non-phage antigens [57]. Increased phage load causes expansion of immune cells in the gut and stimulates IFN-γ secretion through the nucleotide-sensing receptor TLR9, then exacerbates the colon inflammatory response [58]. Phage-caused intestinal inflammation exacerbates the hyperpermeability of the gut.

Anti-T4-like phage antibodies are common in humans, and the capsid proteins Hoc and gp23* contribute significantly to the immune memory of phage T4 [59]. There was evidence that the *B. thetaiotaomicron*, *L. plantarum* and *E. coli* phages induce IFN-γ production in a microbia-independent manner through toll-like receptor 9 (TLR) in the gut [58]. Higher levels of plasma LPS and serum levels of tumor necrosis factor-alpha (TNF-α), IL-1β, and IL-6 were found in rats challenged with the commercial bacteriophage cocktail [44]. Several Toll-like receptors (TLRs), NOD-like receptors (NLRs) and RIG-I-like receptors (RLRs) have been reported as virus PRRs sensors [56]. When phages and bacteria debris meet the intestinal epithelial cells, dendritic cells (DCs) and macrophages within the lamina propria, along with immune cells of gut-associated lymphoid tissue, transcription factors NF-kB, IRF3 and IRF7 are activated through PRRs. Effector molecules such as type I and III interferons (IFN), Interleukin (IL)-6, IL-1β, IL-8 and CXCL-10 are expressed to fight the virus [20,60].

Phages also mediate anti-inflammatory action by eradicating bacterial infection and inhibiting the activity of inflammatory cells during the proinflammatory response [61]. When filamentous Pf phages are taken up by BMDCs, Pf RNA stimulates TLR3-mediated production of type I interferon, which then inhibits TNF production and phagocytosis, thereby inhibiting immunity to *Pseudomonas aeruginosa* infection [62]. Human blood cells treated in vitro, and mice intraperitoneally injected with T4 bacteriophage and head proteins (p23, gp24, Hoc and Soc), exhibited no differences of inflammatory cytokine levels or ROS compared with those of the negative controls [63]. T4 and A3/R bacteriophages had no effect on the differentiation of DCs and the activation of T-cells, but the host bacterial lysate of the phages significantly decreased the percentages of DEC-205- and CD1c-positive cells [64]. T4 bacteriophage could recognize the lipopolysaccharide (LPS) of Escherichia coli by its adhesins (gp12, gp37) [65,66]. The combination decreased the inflammatory response of mice to LPS, as the levels of serum inflammatory markers IL-1α and IL-6, infiltration of leukocytes to lungs, liver and spleen were markedly increased. Phage particles have been found phagocytosed by macrophage in vivo and vitro using fluorescent protein-labeled phage [67,68].

## 4. Phage Therapy for Chronic Liver Diseases

Phage has been reported since 1917 as a treatment for dysentery [69]. This sparked enthusiasm for phages’ treatment of diseases related with specific bacterial pathogens. Over the past two decades, phages have proved effective in fighting antibiotic-resistant bacterial pathogens and the infections they cause [70,71,72,73]. These phages have been successfully used in mice or patients for pathogenic infections such as *Pseudomonas aeruginosa*, *Acinetobacter baumannii*, *Vibrio parahaemolyticus*, *Clostridium difficile*, *Staphylococcus aureus*, *Mycobacterium abscessus* and *Vibrio cholerae* [54,74,75]. Researchers have found that a higher abundance of *Caudovirales* or a higher alpha diversity of bacteriophage in the donor are more effective in treating *C. difficile* infection with fecal bacteria transplantation [76,77].

There have been many clinical trials and case reports of the use of phages in the treatment of gastrointestinal and chronic liver disease [78,79]. Mice receiving phage against adherent-invasive *E. coli* were found to be protected from DSS-induced colitis [80]. A phase I/IIa randomized, double-blind, placebo-controlled clinical trial is underway to assess the safety and efficacy of oral administration of phages that target intestinal adherent-invasive *E. coli* in patients with Crohn’s disease in remission (NCT03808103) [78]. Phages can successfully rescue mice from severe infections caused by multidrug-resistant Klebsiella pneumoniae (ST258) [81]. The use of phage in the treatment of ALD has achieved promising results [15]. Four phages isolated from sewage were able to lyse the intestinal cytolytic E. faecalis strain. Intragastric administration of these phages in *Atp4a^Sl/Sl^* mice, liver damage, steatosis and inflammation induced by chronic ethanol feeding were significantly reduced [15]. Diet-induced obese mice that received faecal virome from lean donors had decreased weight gain, normalized blood glucose and improved bacterial and viral ecosystems [82]. Small intestinal bacterial overgrowth (SIBO) was significantly relieved in high-fat-fed mice that received fecal transplantation of virus-like particles (VLP) compared to control mice [83], while transplanting VLP from high-fat-fed mice into mice on a normal diet simulated high-fat-associated bacterial community composition. Transplantation with active phages reshaped the bacteria of antibiotic-treated mice, very similar to pre-antibiotic microbiota characteristics [84]. Phages could be developed to improve the intestinal microbiome of other patients with chronic liver disease, and thus serve as an effective therapeutic measure.

There are also studies that have shown that phage therapy is ineffective. Several studies of oral use of *E. coli* T4-like phages in adult and child patients with bacterial diarrhea have shown no serious adverse effects, but no efficacy has been observed [85,86]. PreforPro, a phage mixture that targets *E. coli*, was granted in 2017. For patients with mild gastrointestinal distress, there was no difference in the improvement of gastrointestinal discomfort in patients with 28 days of oral PreforPro administration, compared to those with a placebo [87]. When phage was given with *Bifidobacterium lactis*, patients experienced the greatest improvement in gastrointestinal inflammation and pain relief compared to those given *Bifidobacterium lactis* or a placebo alone [88].

Phages are also used as vectors to deliver drugs to specific locations. Drugs are attached to the surface of phage and released when the phage reaches its destination. This technique makes it possible to deliver high doses of drugs to specific sites. Thereby, the concentration of the drug in circulation is reduced and the toxic effect on non-target tissues is reduced. Oral or intravenous injection of irinotecan-carrying phage inhibited the growth of *F. nucleatum* and obviously improved the chemotherapy efficiency of mice with colon cancer [89]. Using their ability to specifically recognize and lyse bacterial isolates, phages can be powerful diagnostic tools for bacterial and even fungal infections [90].

Temperate phages have been underappreciated as a phage treatment because they may not have an immediate bactericidal effect, and show superinfection immunity when integrated with bacterial chromosomes [91,92]. But advances in sequencing and synthetic biology have made it possible for temperate phages to treat bacterial infections [93]. Research is already underway using temperate phages to interfere with the basic metabolic pathways of bacteria to directly kill them, and use temperate phages to make the bacteria less pathogenic or easier to control [93].

## 5. Conclusions

The discovery of links between phages and chronic liver disease has prompted attention to be given to the role of phages in chronic liver disease. By precisely editing the gut microbiome, phages can be a new therapeutic approach against chronic liver disease. Standardized bacteriophage purification, reproducible dosages of bacteriophage and controlled or sustained release at the targeted sites are the keys of phage-personalized therapy. The rapid acquisition of phage resistance by bacteria could disable phages that have already been approved for therapeutic use, which would be a big challenge for us.

## Figures and Tables

**Figure 1 microorganisms-11-01181-f001:**
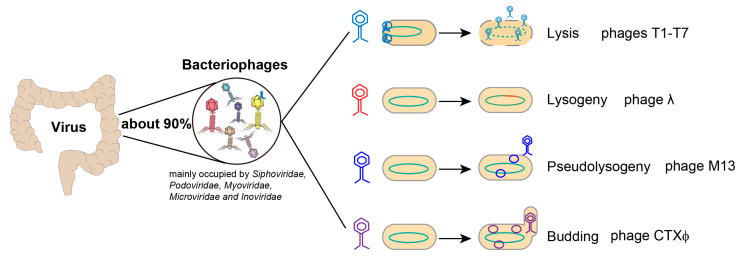
Characteristics of intestinal virus. Bacteriophage is the most abundant component of human intestinal virome (accounting for about 90%). For lytic growth, phages infect bacteria and synthesize their own components, then release new virions by lysing host bacteria. During lysogenic growth, phages inject their genomes into the bacteria and integrate them into the bacterial chromosome. When the phages enter the bacteria body but remain in a static state it is called pseudolysogeny. Finally, some phages infect bacteria and then bud to produce new phages.

**Figure 2 microorganisms-11-01181-f002:**
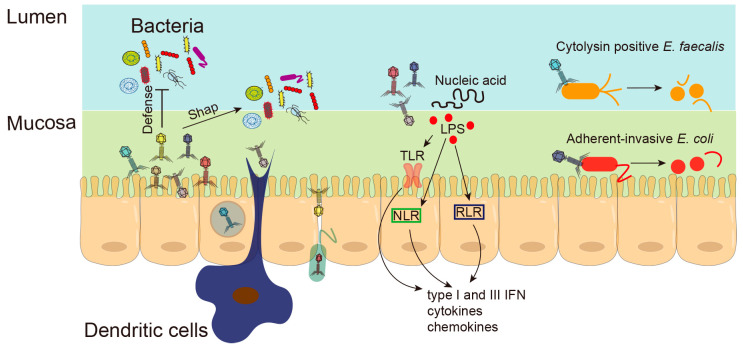
Underlying mechanisms of phages in chronic liver diseases. Phages not only shape the intestinal bacterial colonization, but also prevent bacteria from invading the intestinal barrier. Phages can directly pass through damaged epithelial barriers or be taken up by epithelial cells through transcytosis. Phages hidden in bacteria can cross the epithelial barrier through the Trojan horse mechanism; Phages can also be ingested by enteric dendritic cells. Bacterial components and phage DNA, released from the lysed bacteria, trigger intestinal immunity through Toll-like receptors (TLRs), NOD-like receptors (NLRs) and RIG-I-like receptors (RLRs) in epithelial cells or immune cells. Phages against adherent-invasive *E. coli* were found to be protected from DSS-induced colitis. Phages isolated from sewage were able to lyse the intestinal cytolytic *E. faecalis* strain and alleviated alcohol-induced liver damage.

## Data Availability

Not applicable.

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
