# Peer review of "The Novel Role of Phage Particles in Chronic Liver Diseases"

_microorganisms, 2023, doi:10.3390/microorganisms11051181_

Round 1

Reviewer 1 Report

Chen et al., reviewed the role of phages in liver diseases. Specific comments are listed below

  1. '1. Introduction': the manuscript is in better shape without this section except for lines 48-50. The whole section does not provide information about the rest of the sections and there is no logical flow here.
  2. '2. characteristic of phages': This section can serve as an introduction section better as section '3 alterations in the enteric phages of chronic liver diseases' and '4. Underlying mechanisms of phages in chronic liver disease' is the main section of the manuscript. Change [characteristic] to [characteristics] in the sub-heading
  3. '3. Alterations in the enteric phages of chronic liver diseases and '4. Underlying mechanisms of phages in chronic liver diseases' extensively reviewed phages in the gut or the context of liver diseases but there is lack of discussion about the links/interactions between the phages in the gut and their impact on liver diseases. At least the manuscript is not oriented focusing on this link between gut and liver axes. So it was hard to make a connection as a reader and a reviewer.
  4. 'Figure 1': Ascending colon and descending colon has different microbiome profile. It is not clear here the authors imply that phages are rather associated with descending colon than ascending colon.

Author Response

Chen et al., reviewed the role of phages in liver diseases. Specific comments are listed below

1. '1. Introduction': the manuscript is in better shape without this section except for lines 48-50. The whole section does not provide information about the rest of the sections and there is no logical flow here.

Thank you very much for your advice. We have removed the section except lines 40-68.

2. '2. characteristic of phages': This section can serve as an introduction section better as section '3 alterations in the enteric phages of chronic liver diseases' and '4. Underlying mechanisms of phages in chronic liver disease' is the main section of the manuscript. Change [characteristic] to [characteristics] in the sub-heading

We have moved the section of '2. characteristic of phages' into the introduction according to your suggestions.

3. '3. Alterations in the enteric phages of chronic liver diseases and '4. Underlying mechanisms of phages in chronic liver diseases' extensively reviewed phages in the gut or the context of liver diseases but there is lack of discussion about the links/interactions between the phages in the gut and their impact on liver diseases. At least the manuscript is not oriented focusing on this link between gut and liver axes. So it was hard to make a connection as a reader and a reviewer.

We are agreed with you that the connections of gut and liver were weak. Most studies have shown changes in intestinal phage composition of patients with chronic liver disease, but the causes and consequences of these changes are rarely discussed. We have added descriptions of possible liver-intestinal connections in each section of “Underlying mechanisms of phages in chronic liver diseases”.

4. 'Figure 1': Ascending colon and descending colon has different microbiome profile. It is not clear here the authors imply that phages are rather associated with descending colon than ascending colon.

We are sorry for this misunderstanding, and we have changed the figure 1. Please see the new version.

Reviewer 2 Report

This review by Chen and collaborators focuses on the relationship between the phage roles in the gut microbiota and chronic liver diseases. The review also summarizes the underlying mechanism phages use to shape the bacterial gut microbiota and its metabolism. Finally, the use of phage therapy in chronic liver diseases is also described.

Minor comments

1) Figure1. The authors should consider showing examples of the type of bacteriophages identified in the intestine (i.e., Caudovirales ( Myoviridae, Siphoviridae, Podoviridae) and Microviridae).

2) Section 2. In the text, it will be helpful to provide examples of the four types of phages described in Figure 1, i.e., Virulent phages (Escherichia coli phage T4), Temperate phages (E. coliλ phage)

3) Page 3, lane 103, define MELD as (Model for End-Stage Liver Disease)

Author Response

This review by Chen and collaborators focuses on the relationship between the phage roles in the gut microbiota and chronic liver diseases. The review also summarizes the underlying mechanism phages use to shape the bacterial gut microbiota and its metabolism. Finally, the use of phage therapy in chronic liver diseases is also described.

Minor comments

1) Figure1. The authors should consider showing examples of the type of bacteriophages identified in the intestine (i.e., Caudovirales ( Myoviridae, Siphoviridae, Podoviridae) and Microviridae).

We have added “Siphoviridae, Podoviridae, Myoviridae, Microviridae and Inoviridae” in Figure 1.

2) Section 2. In the text, it will be helpful to provide examples of the four types of phages described in Figure 1, i.e., Virulent phages (Escherichia coli phage T4), Temperate phages (E. coli λ phage)

 We have added examples of the four types of phages in Figure 1.

3) Page 3, lane 103, define MELD as (Model for End-Stage Liver Disease)

 We have added the definition of “MELD”.

Round 2

Reviewer 1 Report

I have no further comments or questions.